# The Influence of Inter-Cooling and Electromagnetic Stirring above Liquidus on the Formation of Primary Al_3_Zr and Grain Refinement in an Al-0.2%Zr Alloy

**DOI:** 10.3390/ma12010022

**Published:** 2018-12-21

**Authors:** Tianyang Guan, Zhifeng Zhang, Yuelong Bai, Min He, Hansen Zheng, Haodong Zhao, Xiaopeng Li, Ping Wang

**Affiliations:** 1General Research Institute for Non-Ferrous Metals, No.2, Xinjiekouwai Street, Xicheng District, Beijing 100088, China; zhangzf@grinm.com (Z.Z.); bai_yuelong@163.com (Y.B.); hemin5209@126.com (M.H.); zhs200559@126.com (H.Z.); zhahadngooo@163.com (H.Z.); lixiaopeng19940501@163.com (X.L.); 2Key Laboratory of Electromagnetic Processing of Materials, Ministry of Education, Northeastern University, No. 3-11, Wenhua Road, Shenyang 110004, China; wping@epm.neu.edu.cn

**Keywords:** grain refinement, inter-cooling, electromagnetic stirring, above liquidus, primary Al_3_Zr

## Abstract

The nucleation and grain growth that occur during solidification have been extensively examined, but insight into the influence of an external field on the formation of heterogeneous crystal nuclei above the liquidus remains unclear in the peritectic refinement mechanism. In this work, we studied the effect of cooling rate above the liquidus on the formation of primary Al_3_Zr and grain refinement in Al-0.2%Zr alloys with inter-cooling annular electromagnetic stirring (IC-AEMS). The results show that the size and distribution of primary Al_3_Zr are greatly improved, and the morphology transformed from large plate/blocky shapes without IC-AEMS to small blocks with IC-AEMS. Meanwhile, above the liquidus, the addition of an Al-Zr master alloy to pure Al alone did little to enhance the refinement, but after IC-AEMS, the grains were refined dramatically. The refinement result seems to be explained by two hypotheses of pre-nucleation and explosive nucleation.

## 1. Introduction

Grain refinement in aluminum and its alloy has always been an important process in production—a fine and equiaxed grain structure can improve the quality of ingots and castings [1]. To achieve a grain-refined structure, two approaches are currently used: the addition of inoculation substances to the alloy (chemical refining) and the application of an external force on the melt (physical refining) [2].

Inoculation and primary intermetallic phases promote heterogeneous nucleation during solidification, including Al-Ti-B, Al-Ti-C, Al-Sc, and Al-Zr master alloys that form precipitates in the melt and act as nucleation sites [3,4,5,6]. Wang et al. revisited the role of peritectic-forming solutes in the grain refinement of aluminum alloys and found that the efficiency of Zr is next to Ti among solutes upon refining [7,8]. However, the size, morphology, and quantity of the nuclei in the refiner are important factors that determine grain refining [9]. Unfortunately, most commercial Refiners note agglomeration or sedimentation during melting and solidification. According to Greer, only 1% of particles succeed in nucleating grains; St. John used interdependence theory to explain why only a small proportion of added inoculant particles are operative [10,11]. The free-growth model can predict the size distribution of inoculant particles on grain size in Al alloys [12]. In general, researchers promote nucleation by adding more refiner during the experiment; however, the solute negatively affects the potency of the nucleants above a certain level [13]. Thus, many studies have been carried out on grain refinement by applying an external field.

Physical refinements can be achieved through various methods, such as bubble mixing [14], electromagnetic stirring [15], and ultrasonic treatment [16]. Several theories have been established to explain the mechanism of grain refining [3]. Gao et al. analyzed the effects of the electromagnetic field and melt treatment near the liquidus on the grain refinement of superalloys—they found that the electromagnetic field could play a positive role in interface stability and dendritic growth to globular transformation [17]. Haghayeghi applied mechanical shearing on the AA7449 aluminum alloy above the liquidus temperature. They found that the shearing temperature had a significant impact on refinement [18]. Eskin et al. studied ultrasonic treatment of the Al-Ti master alloy over different temperature ranges and observed primary Al_3_Ti particles at a quenching temperature above the liquidus temperature after UST [19]. Guan et al. studied the effect of inter-cooling annular electromagnetic stirring (IC-AEMS for short) above the liquidus on the Al-Zn-Mg-Cu alloy with a high elements content, and the solidification structure was greatly refined [20].

Most studies, however, have considered the entire stage of solidification, and have not distinguished the temperature above or below the liquidus in detail. There is still a void in the current understanding of grain refinement above the liquidus—especially the nucleation process. Inoculation and primary intermetallic phases that act as substances or nucleation sites often have higher formation temperatures than the liquidus of the alloy. In binary alloys—especially peritectic reaction alloys such as Al-Zr—the formation temperature of the primary phases is equivalent to the liquidus of the alloy.

In this paper, the Al-0.2%Zr alloy was chosen, and the formation of primary Al_3_Zr and related grain refinements was investigated. The effect of IC-AEMS above the liquidus at a certain cooling rate (3.5 °C/s) was examined, and the possible mechanism and effectiveness of IC-AEMS on grain refinement above the liquidus are discussed in this paper.

## 2. Materials and Methods

### 2.1. Materials and Equipment

The Al-0.2%Zr alloy was prepared using a high-purity commercial aluminum ingot (99.99%) and Al-10Zr master alloy (all the chemical compositions throughout the paper are in weight percent unless otherwise specified). The liquidus temperature of the Al-0.2%Zr alloy was calculated to be 720 °C using the JmatPro software (Sente Software Ltd, Guildford, UK); this was considered the equilibrium precipitation temperature of primary Al3Zr acting as potential nuclei. The chemical composition of the Al-0.2%Zr alloy is listed in Table 1.

The schematic view of the melt treatment apparatus by IC-AEMS is shown in Figure 1a. In this equipment, three-phase alternating currents with a phase angle difference of 120° were imposed on the coils in an electromagnetic stirrer. This led to a rotary magnetic field that subsequently stirred the melt. The inter-cooling rod was used to control the cooling rate of the melt above the liquidus. This space was replenished with cooling air or circulating water through the inlet. This can limit the skin effect area and make full use of the induced current [21]. As shown in Figure 1b, the height of the graphite crucible was 100 mm, the inner radius of the graphite crucible *R* was 40 mm, the outer radius of the cooling rod of the inter-cooling rod *r* was 20 mm, and the thickness of the rod wall was 2 mm. In this work, the frequency of the electromagnetic stirrer was 30 Hz, the stirring current was 60 A, and the pole number was 1.

### 2.2. Melt Treatment Procedure

Pure aluminum was first melted and heated to 850 ± 3 °C in a graphite crucible using an electrical resistance furnace at which point the master alloy was added. The crucible containing the melt was transferred to an electromagnetic stirrer after 30 min of isothermal holding and skimming off the surface oxide skin. A boron nitride-coated cooling rod was preheated to 250–300 °C and inserted in the melt immediately after the cooling air was opened. The melt temperature was monitored by one K-type thermocouple, and it was positioned half the radius away from the wall. The treatment temperature range was from 850 to 720 °C, and the cooling rod was removed when it dropped to 720 °C. The melt without IC-AEMS had a crucible containing the melt placed in the same position, but cooled in air from 850 to 720 °C without electromagnetic stirring and inter-cooling rod insertion.

To compare the efficiency of grain refinement, the melt with and without IC-AEMS above the liquidus was poured into a TP-1 mold (3.5 °C/s) preheated to 350 °C [22]. The melt in the crucible solidified and cooled in the air (0.4 °C/s) to room temperature. This sample was used to investigate the differences between the primary Al_3_Zr phase treated with and without IC-AEMS above the liquidus. The casting conditions, alloy compositions, and their characteristics are summarized in Table 2.

### 2.3. Sample Assessment

Most of the primary Al_3_Zr particles settled to the bottom due to the slow cooling rate (0.4 °C/s) in the crucible and larger density of Al_3_Zr. As shown in Figure 2a, the specimens with and without IC-AEMS were prepared in the central bottom (5 mm from) part of the ingots. To display the 3D morphology of the primary Al_3_Zr particles, the specimens were deeply etched with a 15% NaOH water solution for two hours and then examined with scanning electron microscopy (SEM, JSM-7001F) (JEOL, Tokyo, Japan).

The specimens for grain size assessment were sectioned at a cross-section 38 mm from the base of the TP-1 sample (Figure 2b) and prepared using standard metallographic techniques. The samples were anodized with Barker’s reagent (4% HBF4 in distilled water) and examined under polarized light using the optical microscope (OM, Zeiss Axiovert 200MAT) (Carl Zeiss AG, Heidenheim an der Brenz, Germany). The grain sizes were measured using the linear intercept method (ASTM E112-10).

## 3. Results

### 3.1. The Effect of IC-AEMS on Grain Refinement in Al-0.2%Zr Alloys

To assess the effectiveness of IC-AEMS above the liquidus on grain refinement, pure Al, as well as Al-0.2%Zr with and without IC-AEMS, was poured and solidified in a TP-1 mold at a cooling rate of 3.5 °C/s when the melt temperature dropped to that of the liquidus (720 °C). The as-cast microstructures of pure Al and Al-0.2%Zr alloy samples with and without IC-AEMS above the liquidus are presented in Figure 3. Pure Al has a typical dendrite grain in Figure 3a, but it turns to a columnar grain without IC-AEMS after Al-10%Zr master alloy addition in Figure 3b. The average grain size is reduced from 1383 μm to 797 μm, which indicates that the addition of grain refiner alone above the liquidus can lead to a small refinement efficiency in the Al alloy, to the point of having equiaxed grains—even with high levels of Zr solute. When IC-AEMS is applied, an almost fully refined equiaxed grain structure is obtained in Figure 3c, and the average grain size is reduced from 797 μm to 354 μm. The range of error bars Figure 3d indicates the variation in grain sizes.

### 3.2. The Distribution of Primary Al_3_Zr Particles

Figure 4 presents the micrograph of primary Al_3_Zr particles deposited at the bottom under a slow cooling rate (0.4 °C/s) after being treated with and without IC-AEMS. Two typical primary Al_3_Zr crystals are seen in Figure 4a: several blocky phases and a few rod-like phases in the alloy. There were even a series of Al_3_Zr agglomerates performed without IC-AEMS. In comparison, Figure 4b shows a high density of small, homogeneous particles distributed in the aluminum matrix. The size distribution of the primary Al_3_Zr particles can be well-fitted by a log-normal function, as indicated by the solid line in Figure 4c,d. The maximum diameter observed in the alloy is around 200 μm, while the minimum diameter is just 2 μm in Figure 4c; the mean particle diameter is 51.5 μm without IC-AEMS. After being treated with IC-AEMS, the Al_3_Zr particles are uniformly distributed in the Al matrix, and the mean diameter is greatly decreased to 16.7 μm. Most particles are 2–40 μm (Figure 4d). This shows that the application of IC-AEMS to Al-0.2%Zr alloy above the liquidus refines the primary Al_3_Zr particles.

### 3.3. The Morphology of Primary Al_3_Zr Particles

Figure 5 shows the typical SEM images of the 3D morphology of primary Al_3_Zr particles after deeply etching the samples with and without IC-AEMS. Figure 5a shows that the Al_3_Zr particles without IC-AEMS are generally a plate/blocky-shape with four fast-growing crystallographic directions. However, the morphology of the particles with IC-AEMS was quite different (Figure 5b). There were many small blocky crystals formed in the alloy—especially some tiny erythrocyte-like particles.

## 4. Discussion

In traditional electromagnetic stirring (EMS), when an electromagnetic wave front diffuses into a crucible holding a liquid metal, the “skin effect” will occur and connect with a depth of penetration *δ* [23]:(1)$$\delta =\sqrt{\frac{2}{\sigma \mu \omega }},$$
where *σ* is the electric conductivity; *μ* is the magnetic permeability; and *ω* is the angular frequency, with a value of *ω* = 2*πf*. When the inter-cooling rod is inserted into the melt, the annular gap between the graphite crucible and the rod can make full use of the induced current. When the current frequency f is constant, the induced current density sharply decreases as the penetration depth increases, and the Lorentz force caused by the induced current will change at a different area of the melt. The Lorentz force at the edge is strong, while in the interior, the force is weak. This will cause the heat dissipation to occur faster at the edge than the center. That is the reason why, in traditional electromagnetic stirring (EMS), there is a significant difference in the temperature field and composition field. The average shear rate can be calculated as proposed by Spencer et al. [24]:(2)$$\overset{\cdot }{\gamma }=\frac{2Rr}{R^{2}-r^{2}}\cdot \frac{2\pi n}{60},$$
where $\overset{\cdot }{\gamma }$ is the average shear rate; *R* is the inner radius of the graphite crucible; *r* is the outer radius of the cooling rod; and *n* is the rotation speed of the magnetic field and is determined by [21]:(3)$$n=\frac{60f}{p},$$
where *p* is the pole number; and *f* is the frequency.

In this work, *f* = 30 HZ, *p* = 1, and from Equations (2) and (3), it can be seen that the shear rate $\overset{\cdot }{\gamma }$ is related to the parameter $\left(\frac{r}{R}\right)$. When the stirring frequency is constant, the narrower the gap is, the higher the average shear rate is. For the inter-cooling annular electromagnetic stirring (IC-AEMS), the molten metal is sheared greatly in the annular gap, which is caused by the graphite crucible and the rod. It can avoid the low shear rate area in the center. Moreover, the melt is cooled by the inter-cooling rod, and it can reduce the lateral temperature gradient. Combining the two respects mentioned above, the higher average shear rate can strengthen the stirring of molten metal and heat transmission, which makes the temperature and the distribution of solute elements uniform.

Prior work [12,25,26,27] showed the distribution of a good grain refiner. It should have small particle sizes and no large agglomerates—the shape of the particles reflects the efficiency of the refiners. As shown in Figure 6, the Al-Zr peritectic reaction is different from other foreign nuclei, such as TiB_2_ for Al-Ti-B: the Al_3_Zr particle can completely dissolve in the Al in the alloy above the liquidus. The Al_3_Zr will form as a potential substrate for the nucleation sites of Al grains in the melt once the temperature drops below the liquidus and critical nucleation undercooling is achieved. Thus, this is the initial step to refining grains that control nucleation and growth in Al_3_Zr.

As the melt temperature decreases above the liquidus, the tendency to decrease the Gibbs free energy of the systems drives the active Al atoms and Zr atoms together and stabilizes them in the melt. Figure 7 shows a schematic view of the heat dissipation that might affect the potential nuclei. The measured cooling curves of the Al-0.2%Zr alloy are obvious in Figure 7d. It took about 350 s to reach the liquidus temperature without IC-AEMS; however, it reduced to 50 s with IC-AEMS. Zhu et al. investigated this phenomenon and found that the pouring temperature and cooling rate influence the solidification structure of the Al_3_Zr phase [28,29].

The melt without IC-AEMS has a melt temperature that decreased from 850 to 720 °C under a slow cooling rate. The atoms have sufficient time to migrate and agglomerate; small embryos can be eliminated with fluctuations in the melt, and stable embryos attract more active atoms and grow. The large critical radius is easy to nucleate. The embryos with a small critical radius require more undercooling when the melt temperature falls below the liquidus. The undercooling of growth is smaller than nucleation, and the Al_3_Zr crystals coarsen under the effect of the interfacial energy between the Al melt and the Al_3_Zr particles. The primary Al_3_Zr crystals might consume solute atoms and annex small embryos to grow via Ostwald ripening [30,31]. In summary, the Al_3_Zr particles have size differences and are unevenly distributed across the sample without IC-AEMS.

The IC-AEMS melt has two features that might explain the refining effects of the Al_3_Zr particles. The first one is “pre-nucleation”. It suggests that when the cooling rod is introduced into the melt, the chilling layer around the rod can provide extreme undercooling that leads to instantaneous Al_3_Zr nucleation. However, the forced convection might be due to electromagnetic stirring. This leads to primary Al_3_Zr crystals in the high-temperature area; the crystals melt again as a result. According to the metallogenetic genetic and multi-step nucleation theory [32], the re-melted crystals could retain the stability of the embryos and act as potential nuclei. Similarly, IC-AEMS could increase the nucleation rate.

The second feature was named “Explosive” nucleation. This suggests that the application of inter-cooling could cause the local melt temperature to drop rapidly, but could also affect the electromagnetic field of the overall melt to promote a uniform temperature and composition field [33]. The fluid flow quickly weakens the directional migration movement of the atoms and restricts the growth of the embryos, which have a large radius. The small embryos were activated and dispersed in the melt. These have the same chance of nucleation and thus “Explosive” nucleation occurs. Wang applied high intensity ultrasonic melt treatment to a Al-Ti alloy above the liquidus and also observed refinement of the primary Al_3_Ti intermetallics [19].

The microstructure of the Al-0.2%Zr alloy shown in Figure 3c and the distribution of Al_3_Zr particles presented in Figure 5b clearly indicate that the “Explosive” nucleation mechanism contributes to the refinement of the primary Al_3_Zr particles that act as nucleation sites. The nucleation potency of the small particles plays a dominant role in refinement.

Previous work [34] showed that the growth of the Al_3_Zr crystals is mainly realized by the migration of the lateral interfaces; meanwhile, the lateral surfaces correspond to two families of planes ({101} and {111}) for the smaller plates (Figure 8). The larger sized plates have lateral surfaces that only correspond to the {101} family of planes. The primary Al_3_Zr occurs in the Al-0.2%Zr alloy at 720 °C (Figure 4a). The {101} and {111} have enough time for migration, which allows the particles to grow sufficiently at the lower cooling rate. As the reaction progresses, the {111} planes disappear because of their high roughness [25]. The undercooling degree increased when the IC-AEMS was applied with a cooling rate that increased from 0.4 °C/s to 3.5 °C/s. The critical nuclei diameter is inversely proportional to the melt undercooling and the decreasing Gibbs free energy. Increasing undercooling offers a large driving force whereby more nuclei particles are created. The particles cannot migrate or grow, and thus the {111} planes remain.

## 5. Conclusions


(1)The Al_3_Zr particles have only a minor potency when the alloy is poured at 720 °C; the average grain size is reduced from 1383 μm to 797 μm after addition of the Al-10%Zr master alloy. However, there is significant refinement due to IC-AEMS, with an average grain size that is reduced from 797 μm to 354 μm.(2)IC-AEMS above the liquidus impacts grain refinement due to a reduction in the size of Al_3_Zr particles and their increased density. There is also a more uniform distribution of fine particles in the matrix. The mean particle diameter decreased from 51.5 μm to 16.7 μm, and the morphology of particles transformed from a plate/blocky shape with four fast-growing crystallographic directions to small block-like erythrocyte.(3)The impact of IC-AEMS on grain refinement is attributed to the improved Al_3_Zr precipitates, which act as heterogeneous nuclei in the melt. The use of IC-AEMS further distributes heat and improves the composition above the liquidus. The refinement can be jointly promoted by two hypotheses of pre-nucleation and explosive nucleation.


## Figures and Tables

**Figure 1 materials-12-00022-f001:**
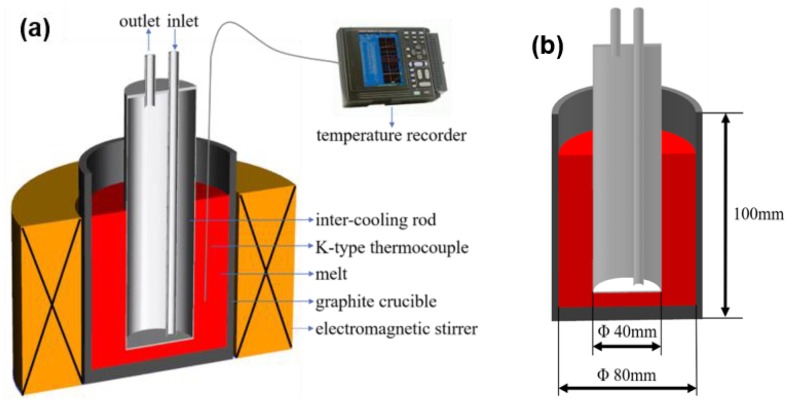
(**a**) Schematic view of the melt treatment apparatus by IC-AEMS, (**b**) the experimental dimensions of the graphite crucible and the inter-cooling rod.

**Figure 2 materials-12-00022-f002:**
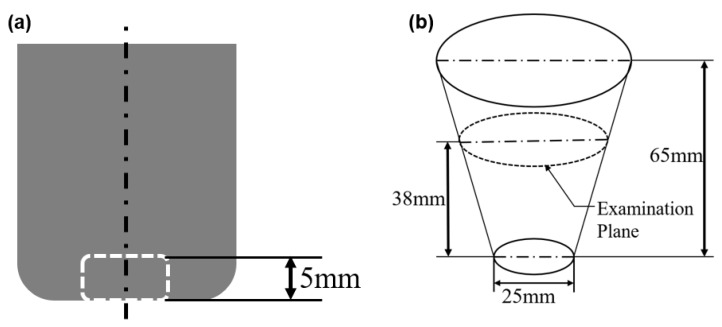
(**a**) Schematic diagram of relative position of specimen at the slow cooling rate (0.4 °C/s); (**b**) schematic diagram of standard TP-1 test specimen.

**Figure 3 materials-12-00022-f003:**
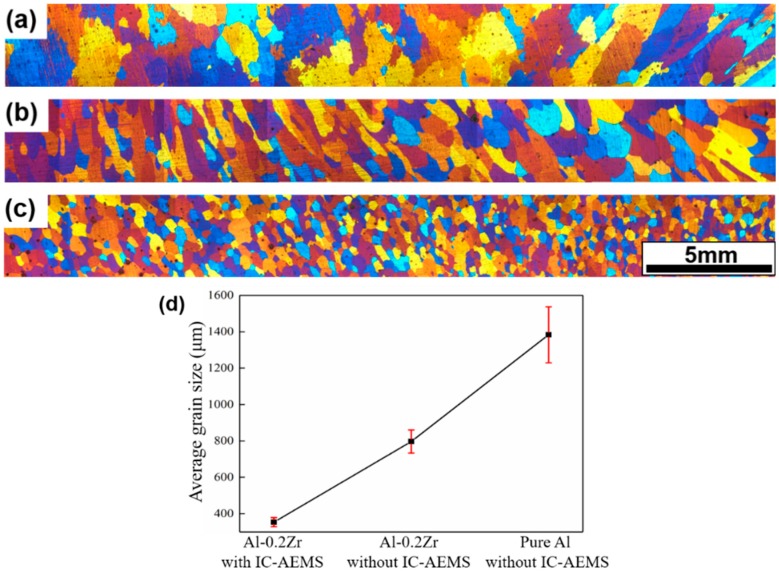
Macrographs showing the grain structures of Al and Al-0.2%Zr poured at 720 °C and solidified in the TP-1 mold (3.5 °C/s). (**a**) Pure Al without IC-AEMS; (**b**) Al-0.2%Zr without IC-AEMS; (**c**) Al-0.2%Zr with IC-AEMS; and (**d**) measured average grain size.

**Figure 4 materials-12-00022-f004:**
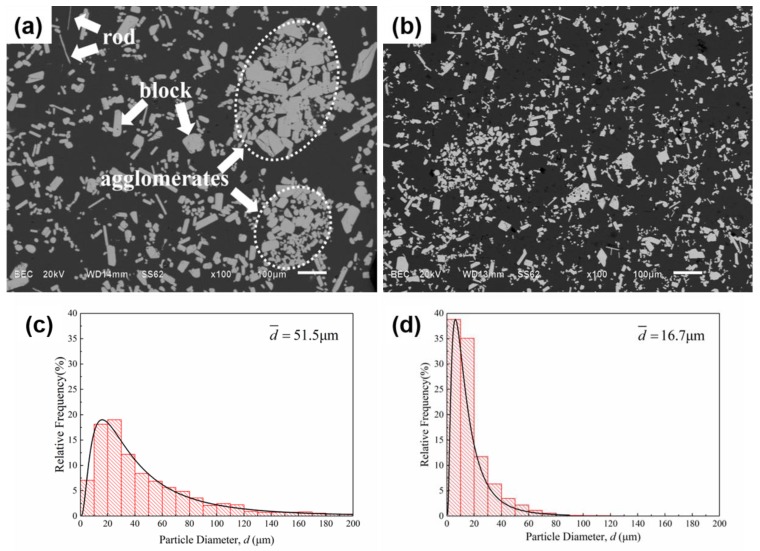
SEM micrographs and size distribution of Al3Zr particles in the Al-0.2%Zr alloy solidified at 0.4 °C/s (**a**,**c**) without IC-AEMS and (**b**,**d**) with IC-AEMS.

**Figure 5 materials-12-00022-f005:**
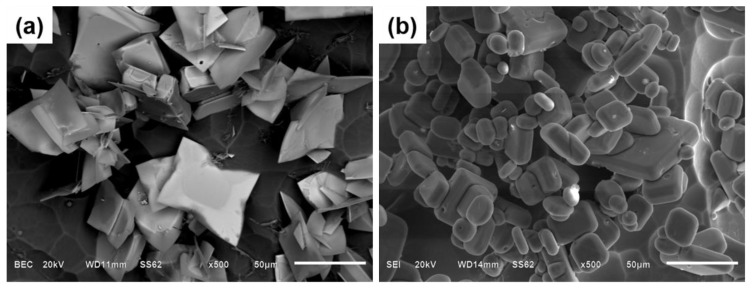
SEM micrographs and diameter distribution of primary Al_3_Zr particles after deeply etching (**a**) without IC-AEMS and (**b**) with IC-AEMS.

**Figure 6 materials-12-00022-f006:**
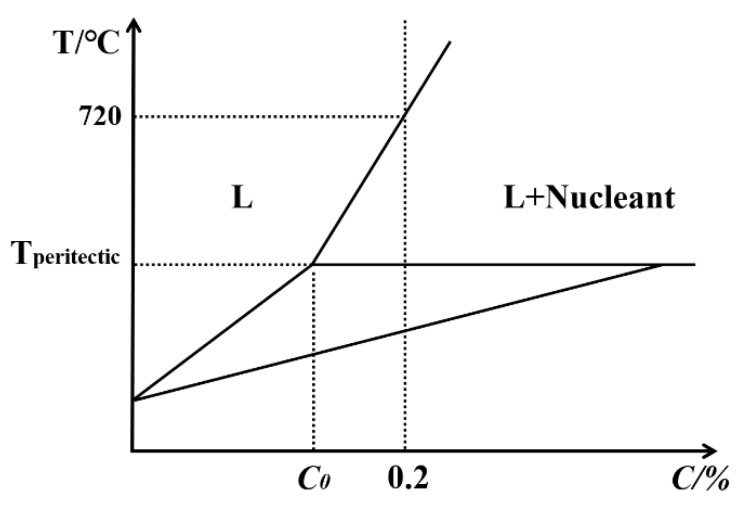
Schematic illustration showing the Al-Zr peritectic reaction.

**Figure 7 materials-12-00022-f007:**
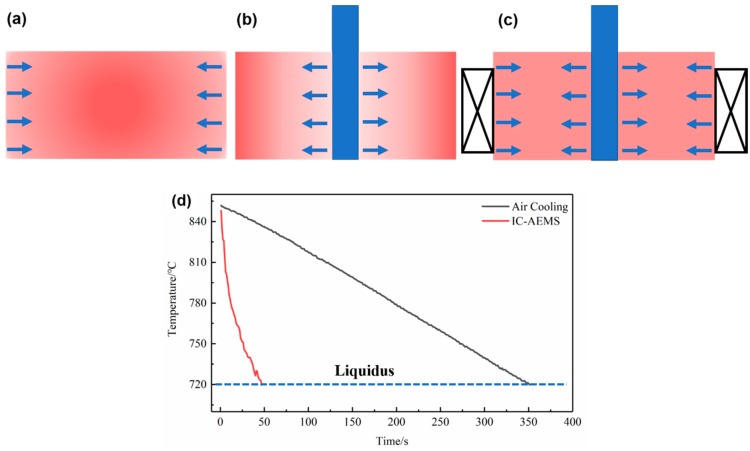
Schematic view of the heat dissipation of the melt in the crucible under different conditions: (**a**) cooling in the air; (**b**) inter-cooling insert; (**c**) inter-cooling with electromagnetic stirring; and (**d**) cooling curves of Al-0.2%Zr alloy with and without IC-AEMS.

**Figure 8 materials-12-00022-f008:**
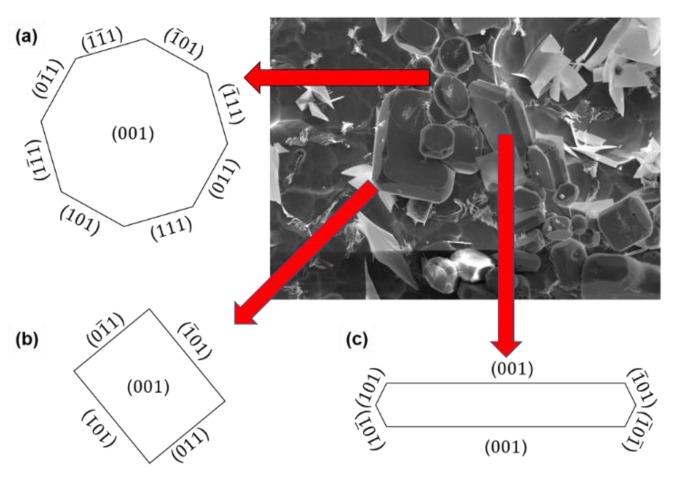
Microstructure of the primary Al_3_Zr crystal in the cross section: (**a**) octagonal form (**b**) and rectangular form; and the longitudinal section: (**c**) tabular form.

**Table 1 materials-12-00022-t001:** Chemical composition of Al-0.2%Zr alloy ((mass fraction, %)).

Zr	Fe	Si	Al
0.217	0.005	0.007	Bal.

**Table 2 materials-12-00022-t002:** Chemical composition, casting conditions, and characteristics of experimental alloys.

Alloy	Temperature of the Phase Formation during Solidification (°C)	Temperature Range of Melt Treatment (°C)	Treatment Condition	Casting Condition	
				Mold	Cooling Rate, °C/s
Al	660(Al)	850–720	-	TP-1	3.5
Al-0.2%Zr	720(Al_3_Zr)		-	TP-1	3.5
Al-0.2%Zr	720(Al_3_Zr)		IC-AEMS	TP-1	3.5
Al-0.2%Zr	720(Al_3_Zr)		-	Graphite crucible	0.4
Al-0.2%Zr	720(Al_3_Zr)		IC-AEMS	Graphite crucible	0.4

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
