# Peer review of "The Influence of Inter-Cooling and Electromagnetic Stirring above Liquidus on the Formation of Primary Al3Zr and Grain Refinement in an Al-0.2%Zr Alloy"

_materials, 2018, doi:10.3390/ma12010022_

Round 1
Reviewer 1 Report
Authors studied the nucleation during solidification of Al-Zr alloy. The paper is well organized and is clearly shown the effectiveness of electromagnetic stirring for grain refinement. Thus, paper should find the high interest to the readers. I have only several minor comments.
1) The fraction of Al3Zr particles seems close to or even higher than that of in Al-0.4Zr [Trans. Nonferrous Met. Soc. China 27(2017) 977−985] despite two times lower Zr content. Did authors observe high homogeneity distribution of Al3Zr or some local areas found higher volume fraction of particles? The maximum fraction of particles can be calculated according to the atomic composition and particles density.
2) What was a composition of pure Al? There were no residual elements? It seems impossible, low amount (0.005 wt%) of residual elements such as Fe and Si are typical even in 99,995 wt% Al grade. This position needs clarification.
3) Several papers [Trans. Nonferrous Met. Soc. China 27(2017) 977−985 Scripta Materialia 133 (2017) 75–78 and Acta Materialia Volume 153, July 2018, Pages 35-44] can be recommended to the authors and can help to improve the introduction and discussion.
Author Response
Dear Reviewer:
Thank you for your careful reading of our manuscript. We have studied comments carefully and have made correction which we hope meet with approval. The main corrections in the paper and the responds to the comments are in the annex.

Reviewer 2 Report
These observations/modifications/specifications that I suggest are the following:
1. There are not indicated (text and/or in Fig. 1) the used experimental dimensions (diameter) of the graphite crucible, of the inter-cooling rod (the proportionality between them) and also the work parameters of the electromagnetic stirrer; these parameters are only theoretically mentioned (R, r, etc.) in eq. (1) -(3);
I suggest to do that in order to corroborate them with dimensions indicated for the sample assessment:
Which part of the entire bottom dimension represent the indicated 5 mm from the center? (line 104)
Also, why exactly 38 mm from the base, used to cut the cross-section? (line 107); what is the proportion with the entire ingot high?
2. Related to above obs. 1, how were correlated the real applied experimental parameters (R, r, f, d etc.) to theoretically mentioned ones from eq. (1)-(3) and, most of all, to discussion from lines 170-172? There is a comment related to theoretical equations, but what about the real experimental conditions? Please, provide an explanation in that direction.
3. line 166: is missing the reference number for “Spencer et al”, and the reference it-self in the final list. In plus, the eq. (1) -(3) must be accompanied by a reference source also. Please, fix the problem.
4. Fig. 5, with peritectic reaction, is not mentioned in the text. Please, add a reference to this figure in the text.
Author Response

(The authors gave the same response as above.)
